# A high latitude Gondwanan species of the Late Devonian tristichopterid *Hyneria* (Osteichthyes: Sarcopterygii)

**Robert W. Gess**[1☯]*, **Per E. Ahlberg**[2☯]*

**1** Albany Museum and Geology Department, Rhodes University, Makhanda/Grahamstown, South Africa,
**2** Department of Organismal Biology, Uppsala University, Uppsala, Sweden

☯ These authors contributed equally to this work.
* robg@imaginet.co.za (RWG); per.ahlberg@ebc.uu.se (PEA)

## Abstract

We describe the largest bony fish in the Late Devonian (late Famennian) fossil assemblage from Waterloo Farm near Makhanda/Grahamstown, South Africa. It is a giant member of the extinct clade Tristichopteridae (Sarcopterygii: Tetrapodomorpha) and most closely resembles *Hyneria lindae* from the late Famennian Catskill Formation of Pennsylvania, USA. Notwithstanding the overall similarity, it can be distinguished from *H. lindae* on a number of morphological points and is accordingly described as a new species, *H. udlezinye* sp. nov. The preserved material comprises most of the dermal skull, lower jaw, gill cover and shoulder girdle. The cranial endoskeleton appears to have been unossified and is not preserved, apart from a fragment of the hyoid arch adhering to a subopercular, but the postcranial endoskeleton is represented by an ulnare, some semi-articulated neural spines, and the basal plate of a median fin. The discovery of *H. udlezinye* shows that *Hyneria* is a cosmopolitan genus extending into the high latitudes of Gondwana, not a Euramerican endemic. It supports the contention that the derived clade of giant tristichopterids, which alongside *Hyneria* includes such genera as *Eusthenodon*, *Edenopteron* and *Mandageria*, originated in Gondwana.

## Introduction

Tristichopterid fishes represent the sister group of elpistostegalians and digit bearing tetrapods. They achieved a worldwide distribution during the later part of the Devonian Period, before becoming extinct during the End Devonian Mass Extinction Event [1]. Most known taxa have either been recovered from the tropical to subtropically deposited sediments of Euramerica or alternately from Australia which, towards the end of the Devonian, formed the low latitude northern rim of Gondwana. Olive *et al.* [2] have hypothesised a western European (Euramerican) origin for tristichopterids, suggesting that they migrated from Western Europe to Australia as part of the Late Devonian faunal exchange between Euramerica and Gondwana. Thereafter they suggest that a highly nested clade of giant Late Devonian tristichopterids, comprising *Cabonnichthys burnsi* [3], *Mandageria fairfaxi* [4], *Eusthenodon wangsjoi* [5],

**Data Availability Statement:** All relevant data are within the manuscript and its Supporting information files. In addition, all specimens described in the manuscript are held by the Albany

Museum, Makhanda/Grahamstown, South Africa, and are available for study.

**Funding:** PEA: Wallenberg Scholarship (not numbered), from the Knut & Alice Wallenberg Foundation. https://kaw.wallenberg.org PEA: ERC Advanced Grant ERC-2020-ADG 10101963 "Tetrapod Origin", from the European Research Council. https://erc.europa.eu/homepage RWG: Millennium Trust, South Africa (no number). https://www.mtrust.co.za RWG: GENUS (DSI-NRF Centre of Excellence in Palaeosciences), South Africa (no number). https://www.genus.africa RWG: NRF of South Africa (no number). https://www.nrf.ac.za The funders had no role in study design, data collection and analysis, decision to publish, or preparation of the manuscript.

**Competing interests:** The authors have declared that no competing interests exist.

*Edenopteron keithcrooki* [6], and *Hyneria lindae* [7] most likely differentiated in Australia. Of these *Cabonnichthys*, *Mandageria* and *Edenopteron* are known from single localities in eastern Australia, whereas *Eusthenodon* is known from Greenland, Western Europe, Russia and Australia, and *Hyneria* is known from Pennsylvania in North America [2, 8, 9]. *Eusthenodon* and *Hyneria* are hypothesised to have migrated to Euramerica from an Australian centre of origination during the Famennian [2].

The greatest limitation of this biogeographical scenario is that in designating Australia specifically as the centre of endemism of the proposed Gondwanan originated clade, it overlooks the possible effects of collection bias. Although Gondwana was the largest landmass during the Devonian, extending from tropical to polar regions, investigation of Late Devonian early vertebrate faunas has largely been restricted to Australia and palaeo-adjacent portions of Antarctica. Here we describe a new species of *Hyneria* from the Gondwanan high-latitude lagerstätte at Waterloo Farm, South Africa (Fig 1). This has important implications for our understanding of the biogeographical distribution and habitat preferences of Late Devonian tristichopterids.

## Methods

### Material and techniques

The fossil material was excavated from a roadside cutting outside Makhanda/Grahamstown and is now housed at the Albany Museum in that city. Specimens were prepared mechanically by hand, without the use of power tools; the soft sediment was in most cases slowly flaked off with a porcupine quill in order to protect the fragile and diagenetically altered bone compressions. The bones were photographed without any form of coating. Interpretative drawings were traced directly from the photographs, in Adobe Photoshop, with the specimens to hand for checking. Where possible, information was combined from the part and counterpart of the bone. The three-dimensional skull reconstruction was produced by fitting together paper outlines of the bones to make a physical three-dimensional model, which was then photographed and traced in Adobe Photoshop. All material was collected by the first author, with the

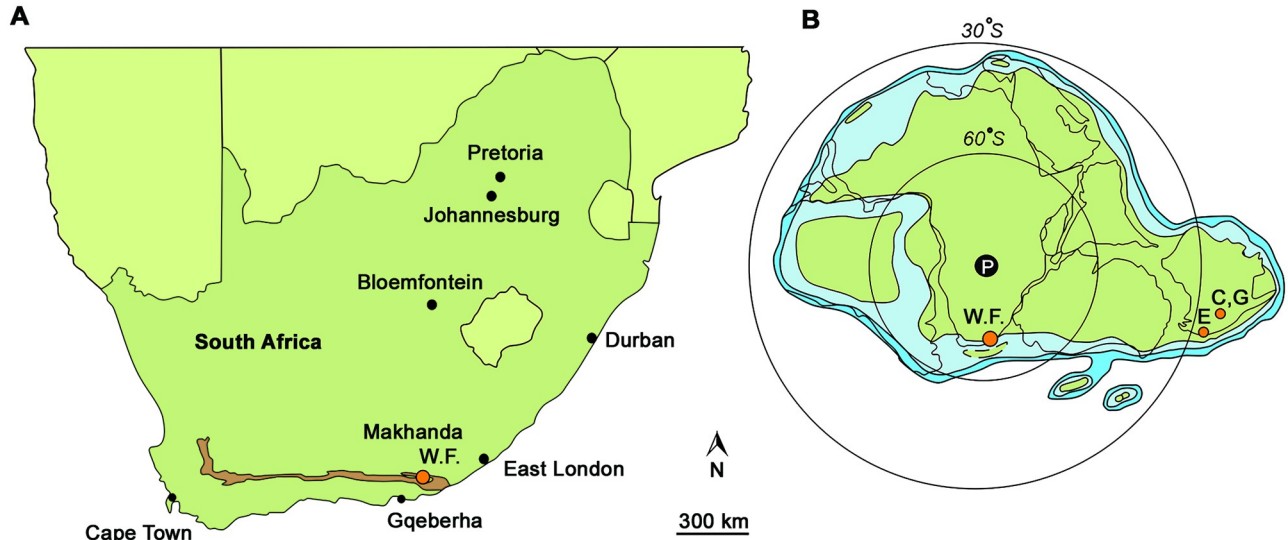

**Fig 1. The location of Waterloo Farm.** A, map of South Africa showing position of Waterloo Farm (W.F.). Brown area denotes the outcrop of the Witpoort Formation. B, palaeogeographical map of Gondwana in south polar view during the Famennian, showing the position of Waterloo Farm (W. F.) and three other Late Devonian tristichopterid localities, Eden (E), Canowindra (C) and Grenfell (G), relative to the South Pole (P). A modified after [10], B modified after [11].

permission of SANRAL (the South African National Roads Agency; land owner) and under permits granted by SAHRA (the South African Heritage Resource Agency) and ECPHRA (the Eastern Cape Heritage Resource Agency), before being accessioned into the Devonian collection of the Albany Museum, on which the first author is the principal researcher. One exception is specimen AM18000, collected by Ryan Nel under the first author's supervision. No permits were required for the described study, which complied with all relevant regulations. Permits were not required because the first author is employed by the Albany Museum specifically to curate and conduct research on their Devonian collection, of which this material forms a part. The second author was invited by the first author to collaborate on analysis of the material.

### Nomenclatural acts

The electronic edition of this article conforms to the requirements of the amended International Code of Zoological Nomenclature, and hence the new names contained herein are available under that Code from the electronic edition of this article. This published work and the nomenclatural acts it contains have been registered in ZooBank, the online registration system for the ICZN. The ZooBank LSIDs (Life Science Identifiers) can be resolved and the associated information viewed through any standard web browser by appending the LSID to the prefix "http://zoobank.org/". The LSID for this publication is: urn:lsid:zoobank.org: pub: 3112D493-2967-448F-B095-52892107DDC1. The electronic edition of this work was published in a journal with an ISSN, and has been archived and is available from the following digital repositories: PubMed Central, LOCKSS, DiVA (Uppsala University repository).

## Results

### Systematic palaeontology

OSTEICHTHYES Huxley, 1880

 SARCOPTERYGII Romer, 1955

 TETRAPODOMORPHA Ahlberg, 1989

 TRISTICHOPTERIDAE Cope, 1889

 *Diagnosis*—Tetrapodomorph sarcopterygians with postspiracular bone present, vomers with long caudal process clasping the parasphenoid, circular scales with a median boss, and an elongate body with a trifurcate or rhombic caudal fin (modified from [3]).

 *HYNERIA* Thomson, 1968

 *Type species*—*Hyneria lindae* Thomson, 1968; Hyner, Pennsylvania, USA.

 *Hyneria udlezinye* sp. nov. urn:lsid:zoobank.org:act: B05B6F7E-AF8C-4169-B992-24F7FFF59EE0

 "Probable eusthenopterid" [12]

 "Close to *Eusthenodon*" [13]

 "Similar to *Hyneria*" [14]

 "cf *Hyneria*" [15]

 "*Hyneria*-like" [16]

 *Diagnosis*—A very large tristichopterid, closely resembling *Hyneria lindae* but differing from it in the following respects: postparietal shield widening more strongly from anterior to posterior; lateral corner of tabular weakly developed; preopercular and lacrimal proportionally deeper; denticulated field on parasphenoid extends further anteriorly; subopercular more shallow; dentary fangs proportionately larger.

 *Etymology*—an apposition, from isiXhosa 'udlezinye', meaning 'one who eats others', referring to the inferred predatory lifestyle of the species. IsiXhosa is the widely spoken indigenous language of south-eastern South Africa where the fossil locality is located.

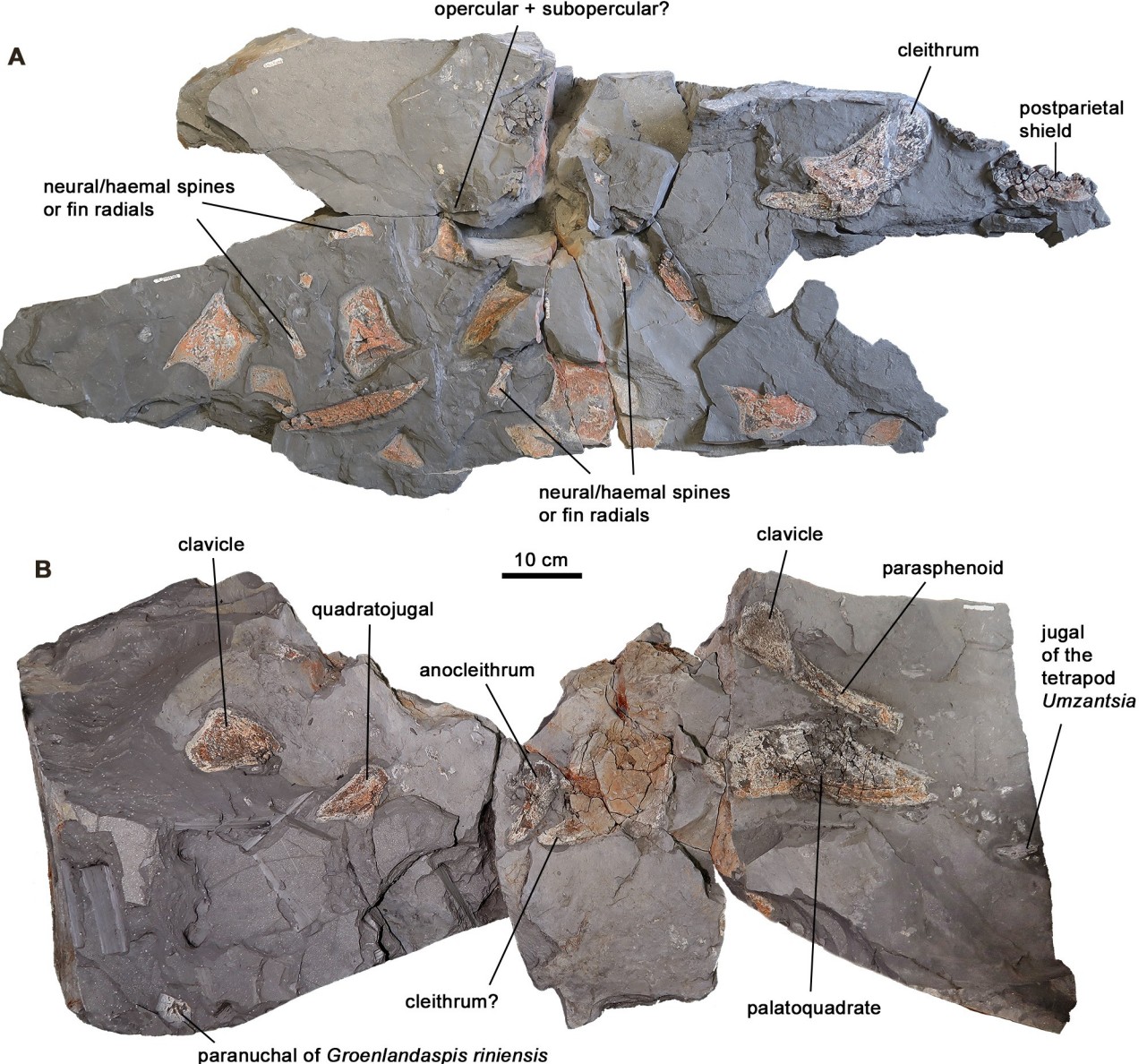

**Fig 2. AM6540b and AM6528a, the two main blocks of the holotype of *Hyneria udlezinye*.** Each block also has a counterpart (not illustrated). A, AM6540b. Unlabelled bones all belong to a single large individual of the arthrodire placoderm *Groenlandaspis riniensis* [19]. B, AM6528a. This block also carries a jugal of the tetrapod *Umzantsia amazana* [11] and a paranuchal of a small individual of *Groenlandaspis riniensis*.

*Holotype*—AM6528 + AM6540 (Fig 2) + AM5532 (Fig 8).

*Referred material*—AM4868, AM5249, AM5389, AM5391, AM5532, AM5668, AM5888, AM6502, AM6503, AM6508, AM6516, AM6521, AM6526, AM6533, AM6535, AM6541, AM6542, AM6545, AM18000, AM18001, AM18002, AM18003, AM18004. All material is housed in the Albany Museum, Makhanda/Grahamstown, South Africa.

*Locality and age*—Waterloo Farm near Makhanda/Grahamstown, South Africa. Witpoort Formation; Late Devonian, late Famennian.

## Description

The *Hyneria* material from Waterloo Farm consists predominantly of dermal bones, though some elements of the axial skeleton and paired fins are also preserved (Fig 2). Material is preserved in muddy carbonaceous metashale. During metamorphosis thicker bone became internally carbonized, though its outer surface is defined by kaolin derived from breakdown of secondary mica. The latter style of preservation also comprises thinner elements such as scales. The large size and robustness of the bones means that the dermal ornament and overlap surfaces are, for the most part, clearly visible. The suite of dermal skull bones is complete enough to allow a well-constrained reconstruction of the cranial morphology (Fig 3). Lateral line canals are generally not well preserved because of the compaction of the material, but can be traced in some areas. Curiously, the teeth are much less well preserved than the bone tissue; they tend to be black rather than silvery in colour, and even the largest of them (such as the dentary fang pair) do not show complete natural outlines. No cephalic endoskeletal elements of *Hyneria* have been recovered from Waterloo Farm. This may be due to preservational factors, but *Hyneria* appears in any case to be characterised by a lightly ossified endoskeleton. The type species *Hyneria lindae* from the Catskill Formation, Pennsylvania, has an ossified Meckelian element, palatoquadrate, hyomandibula, scapulocoracoid and pectoral fin skeleton, but the braincase seems to be completely unossified [17, 18]. Some remains of the axial and appendicular skeleton of *Hyneria udlezinye* have been recovered, including, generally disassociated, neural and haemal arches, a fin support, lepidotrichia and an ulnare.

Because *Hyneria udlezinye* is in most respects extremely similar to the type species *Hyneria lindae*, with only proportional differences in some bones serving to distinguish the two species, the Waterloo Farm material will be described relatively briefly and each element in turn will be compared with that of *H. lindae*. Comparisons with other tristichopterids will be made in the discussion. The distribution of bones on the bedding planes and their relative sizes allow us to assign certain suites of bones to single individuals with reasonable confidence. AM6528 + 6532 + 6540, which we designate as the holotype of *Hyneria udlezinye*, comprises an incomplete postparietal shield, quadratojugal, incomplete opercular, anocleithrum, cleithrum and clavicle. AM6535 comprises an incomplete parietal shield and a preopercular, AM6542 a postorbital and a lateral extrascapular. It has not proved possible to assign the remaining bones with confidence to these individual assemblies, though some almost certainly belong to them.

**Skull roof.**   The parietal shield is represented by a single almost complete right parietal with an attached complete intertemporal (Fig 4A–4D). Well-defined overlap areas for the median rostral, nasal and anterior supraorbital (or prefrontal in tetrapod terminology) are present along its anterior and anterolateral margins. The contact for the posterior supraorbital (postfrontal) is a broad underlap on the ventral surface of the parietal; this feature can also be seen in *Hyneria lindae*, although it is not labelled [17: fig 3A]. The medial margin of the parietal carries a notch for the pineal region, which appears to have been drop-shaped with a posterior point. The anterior part of the main trunk of the supraorbital lateral line is visible on the dorsal surface of the parietal, along with two smaller anterolaterally directed side branches. The intertemporal carries a large lateral overlap flange for the postorbital.

The proportions of the parietal shield are distinctive, with a notably short and broad postpineal region (Fig 3A). This feature is shared with *H. lindae*, and possibly with *Langlieria socqueti* [20], but not with other derived tristichopterids such as *Eusthenodon*, *Mandageria*, *Cabonnichthys* or *Edenopteron* [3–6, 21], in which this region is proportionately longer and often tapers posteriorly towards a proportionately shorter posterior margin.

The postparietal shield is represented by a single incomplete postparietal together with the associated tabular and supratemporal (Fig 4D and 4E). The two latter bones are complete, but

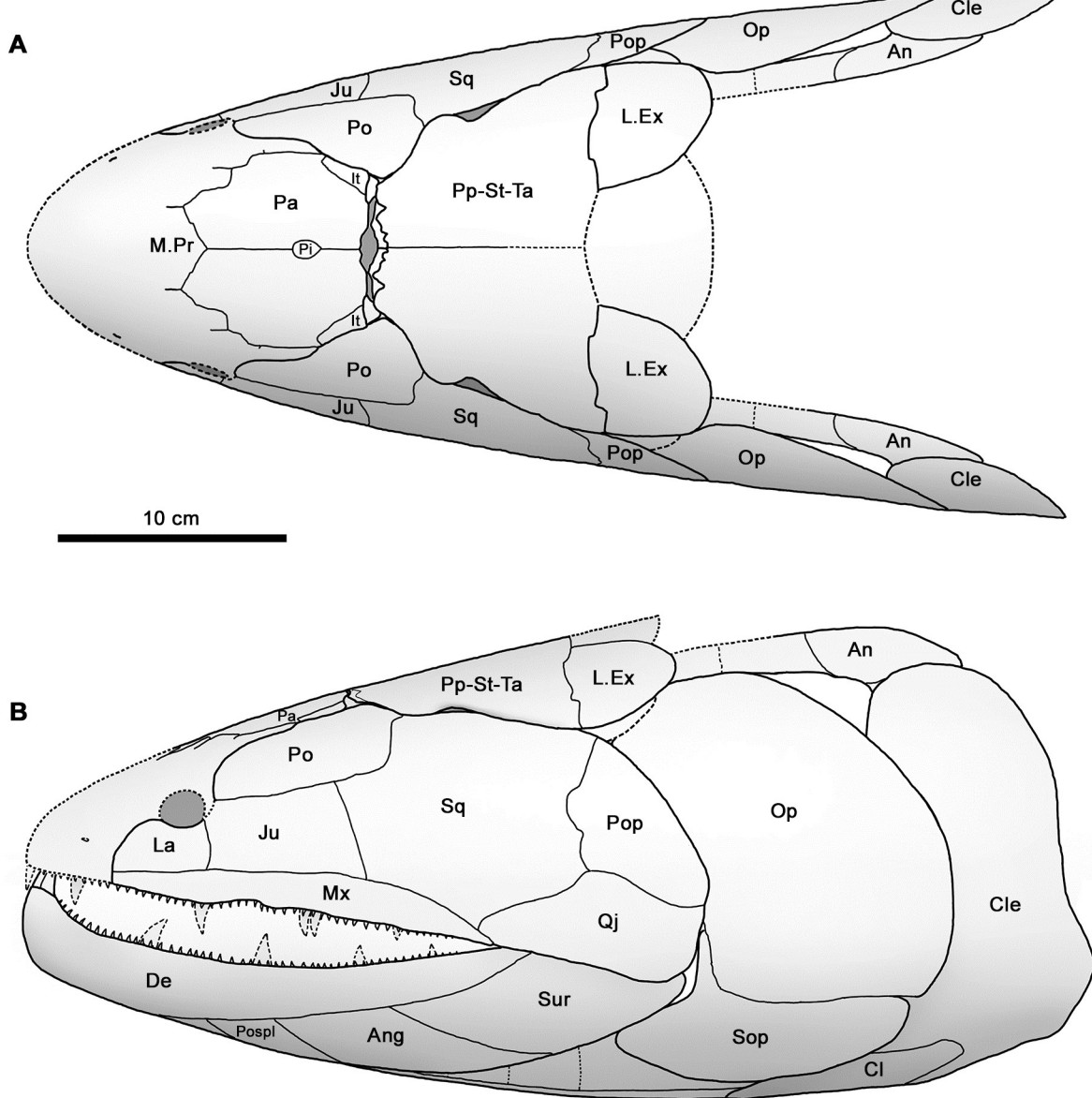

**Fig 3. Skull reconstruction of *Hyneria udlezinye*.** Dorsal (A) and lateral (B) views, drawn from photographs of a three-dimensional model, scaled to the size of the holotype. Abbreviations: An, anocleithrum; Ang, angular; Cl, clavicle; Cle, cleithrum; De, dentary; It, intertemporal; Ju, jugal; La, lacrimal; M.Pr, median postrostral; Mx, maxilla; Op, opercular; Pa, parietal; Pi, pineal; Po, postorbital; Pop, preopercular; Pospl, postsplenial; Pp-St-Ta, postparietal, supratemporal and tabular (sutures not visible); Qj, qudratojugal; Sop, subopercular; Sq, squamosal; Sur, surangular.

their sutures with the postparietal cannot be traced. A short section of the postotic lateral line canal is visible in the posterior part of the tabular. The outline of the postparietal shield differs from that of *H. lindae* in the smoother curve of the lateral tabular margin, which does not have a distinct lateral corner and is slightly longer with a more posteriorly positioned posterolateral corner. The anterior margin of the shield (i.e. the dermal intracranial joint) is also proportionately shorter than in *H. lindae* and the contact margin for the postorbital on the supratemporal correspondingly more oblique.

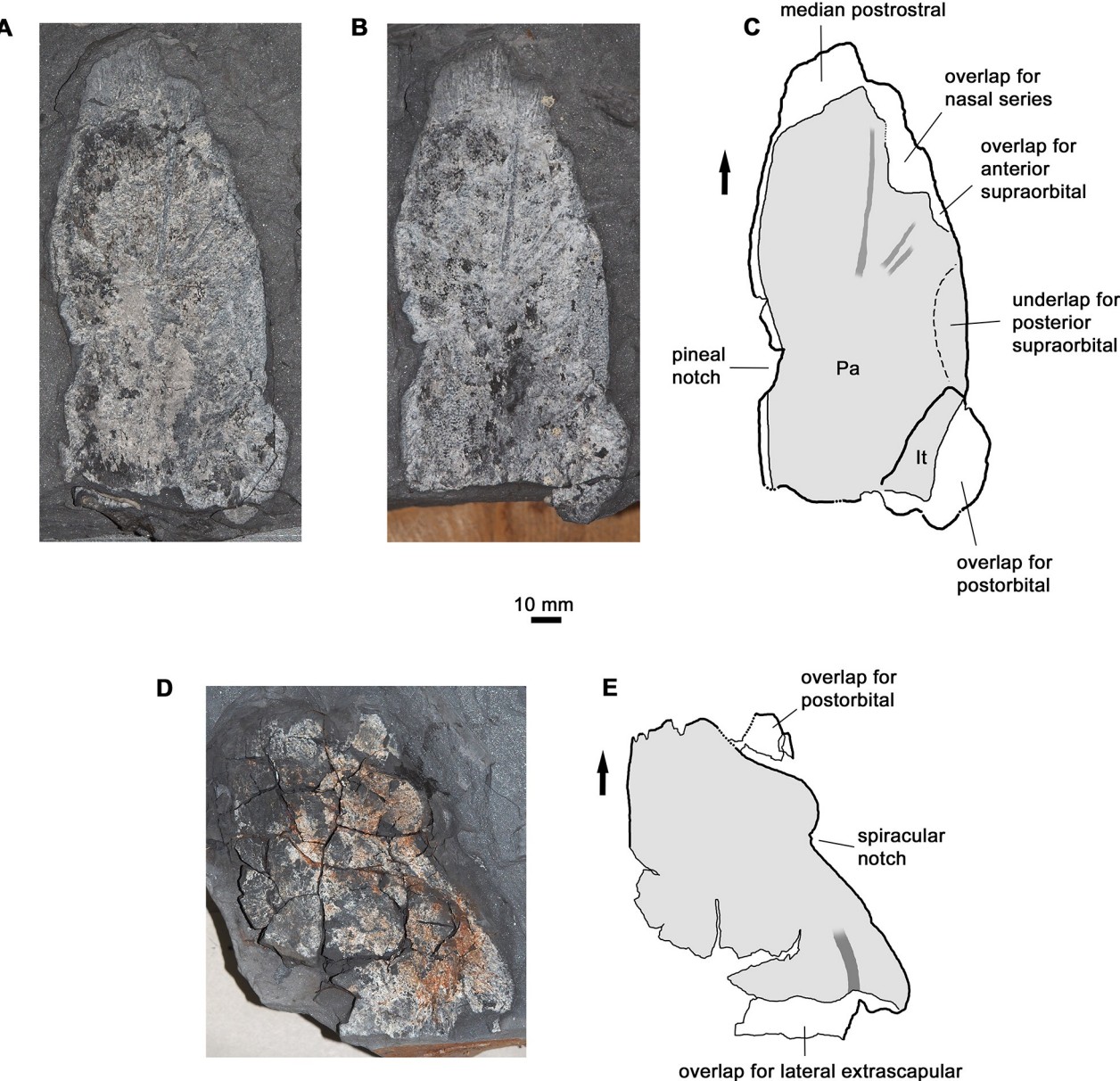

**Fig 4. Skull roof bones.** A-B, specimen photographs of parietal plus intertemporal, AM6535c,d, in part and counterpart (left-right reversed) view. C, interpretative drawing combining information from A and B. D, specimen photograph of postparietal-supratemporal-tabular (sutures not visible), AM6540a. E, interpretation of D. Thick outline denotes true margin; dotted thick outline, inferred true margin; thin outline, broken margin or edge of ornamented external surface; pale grey, ornamented external surface; dark grey, lateral line canal; black arrow, anterior. Abbreviations as in Fig 3. Specimens are shown to the same scale but the parietal represents a larger individual.

**Cheek.** The cheek is represented by the postorbital, preopercular, quadratojugal, squamosal, maxilla and lacrimal (Fig 5); no jugal has been discovered in the material. The postorbital has a characteristic tristichopterid shape, an elongated triangle with a posterior margin that terminates in a posterodorsal process. The overlap surface for the posterior supraorbital (post-frontal) is well preserved and covers the entire anterodorsal corner of the bone. The postorbital was thus excluded from the orbit by a ventral process of the posterior supraorbital that reached

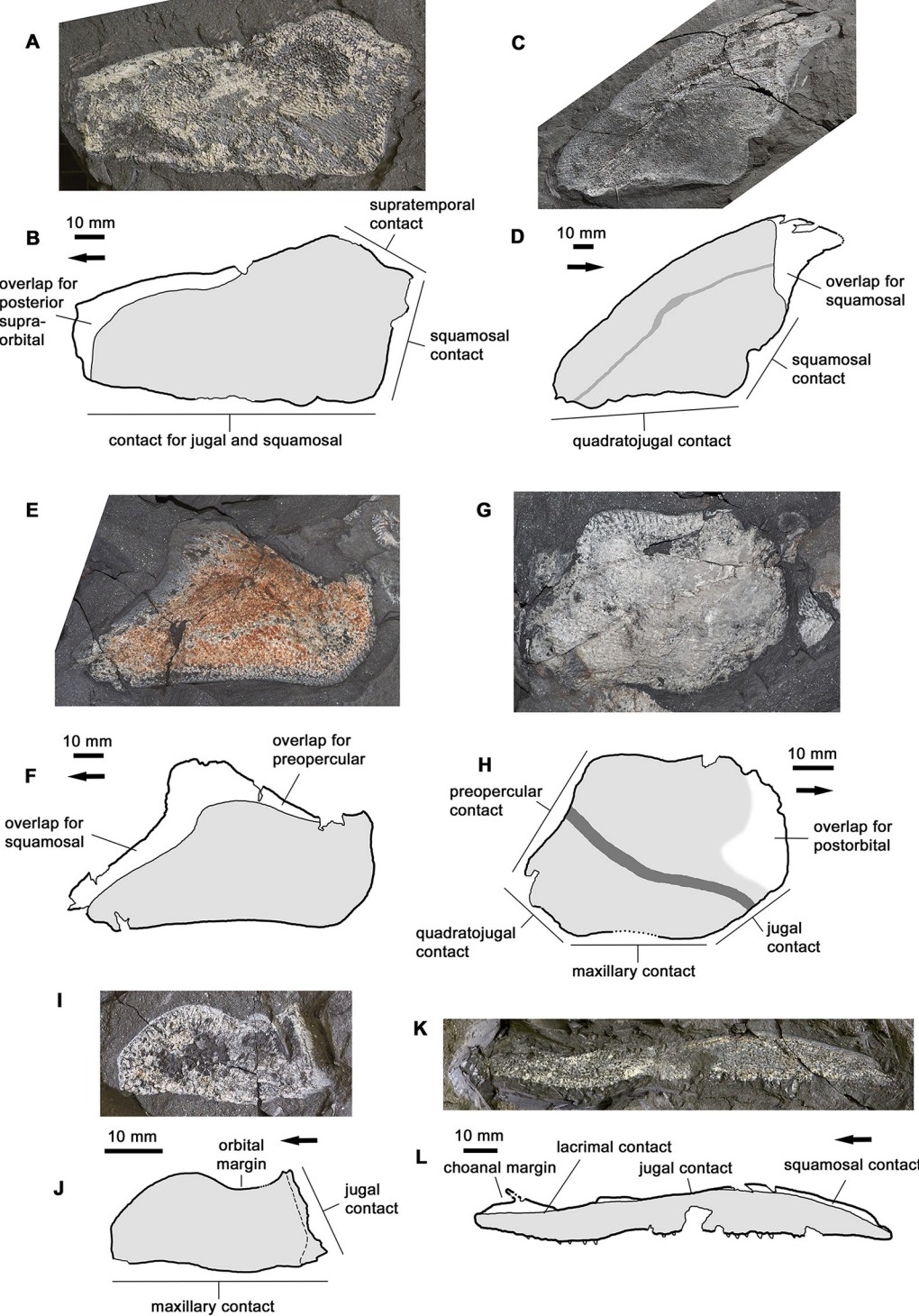

**Fig 5. Cheek bones.** A-B, photo and interpretative drawing of postorbital AM6541b. C-D, photo and interpretative drawing of preopercular AM6535b. E-F, photo and interpretative drawing of quadratojugal AM6528b. G-H, photo and interpretative drawing of squamosal AM18001. I-J, photo and interpretative drawing of lacrimal AM6526b. K-L, photo and interpretative drawing of maxilla AM6503a. Graphic conventions as in Fig 4. In H, the edge of the postorbital overlap is indistinct and is shown as a blurred margin.

down to contact the jugal. This condition is typical for the clade of derived tristichopterids that includes *Hyneria*, *Eusthenodon*, *Edenopteron*, *Mandageria* and *Cabonnichthys* [2, 3].

The maxilla is almost parallel-sided. Again, this is typical for the aforementioned clade of derived tristichopterids, and differs from the condition seen in *Eusthenopteron* and almost all non-tristichopterid 'osteolepiforms', where the maxilla has a distinct dorsal process peaking at the jugal-squamosal junction [22, 23]. The maxilla of *Hyneria lindae* is even more extreme than that of *H. udlezinye*, in that the contact margin for the squamosal is concave rather than convex [17: fig 5]. A parallel-sided maxilla is also present in tetrapods, although this is probably convergent with the derived tristichopterid condition. Near the anterior end of the dorsal margin, the maxilla carries a slender anterodorsally directed process. This process contributes to the posterior margin of the choana and is seen in many tetrapodomorph fishes [17, 23].

The remaining bones of the cheek are of generalized tristichopterid character. Comparison with *Hyneria lindae* is made somewhat more difficult by the fact that the articulated *H. lindae* cheek-plate figured by Daeschler & Downs [17] is shown only in internal view, whereas the *H. udlezinye* cheek bones are exposed in external view. Tristichopterid cheekbones tend to have large overlap surfaces [22: fig 115], meaning that the internal and external suture patterns can look rather different. In the *H. udlezinye* material we note particularly that the squamosal has a very extensive overlap surface for the postorbital (Fig 5G and 5H). Nevertheless, comparison between the two species indicates that *H. udlezinye* differs from *H. lindae* in having a proportionately deeper preopercular (in *H. lindae* this bone is very narrow and bar-shaped) and a shorter, deeper, lacrimal. In both these respects, *Hyneria udlezinye* appears to be the more generalized of the two, matching the condition in a wide range of tristichopterids including *Eusthenopteron*, *Cabonnichthys*, *Eusthenodon* and *Edenopteron* [3, 5, 6, 21, 22].

**Lower jaw.** The lower jaw is poorly represented in the material. Only the dentary and an incomplete angular are known as isolated bones (Fig 6A, 6B, 6D and 6E). An articulated but partly disrupted lower jaw ramus in mesial view shows the prearticular in association with some of the smaller bones (Fig 6C) but is in some respects challenging to interpret. The angular is of a generic tristichopterid type, but the dentary has a distinctive, almost club-shaped morphology with a deep anterior end carrying the poorly preserved remains of a large fang pair (Fig 6A and 6B). Comparison with the lower jaw of *Hyneria lindae*, which unfortunately is figured only in mesial view so that the complete outline of the dentary cannot be seen, shows that the dentary fang pair of *H. lindae* is only about half the relative size of that of *H. udlezinye*, and suggests that the dentary of *H. lindae* is shallower at the anterior end.

**Palate.** The palate of *Hyneria udlezinye* is represented by several entopterygoids of varying degrees of completeness and a complete, partially exposed parasphenoid (Fig 7). The left entopterygoid of the holotype individual is almost complete when information from the part and counterpart is combined (Fig 7B). In addition to the main corpus of the bone, which was exposed on the palate in life, the entopterygoid also has an extensive ventral flange that formed the attachment area for the ectopterygoid and dermopalatine. The two latter bones were thus not in direct contact with the palatoquadrate. The same arrangement was documented by Jarvik [22: fig 108] in the grinding series of *Eusthenopteron*. As the palatoquadrate complex of *H. lindae* (which includes entopterygoid, dermopalatine and palatoquadrate) is incomplete posteriorly, it is somewhat difficult to compare with the entopterygoid of *H. udlezinye*. However, in *H. udlezinye* the denticulated field has a more pronounced dorsal corner and extends closer to the dorsal margin of the bone in the metapterygoid region.

The parasphenoid, which also belongs to the holotype individual, is exposed in right ventrolateral view (Fig 7E and 7F). Like the corresponding bone in *H. lindae* it has a very long overlap area for the vomer, but the denticulated field extends further anteriorly than in *H. lindae*.

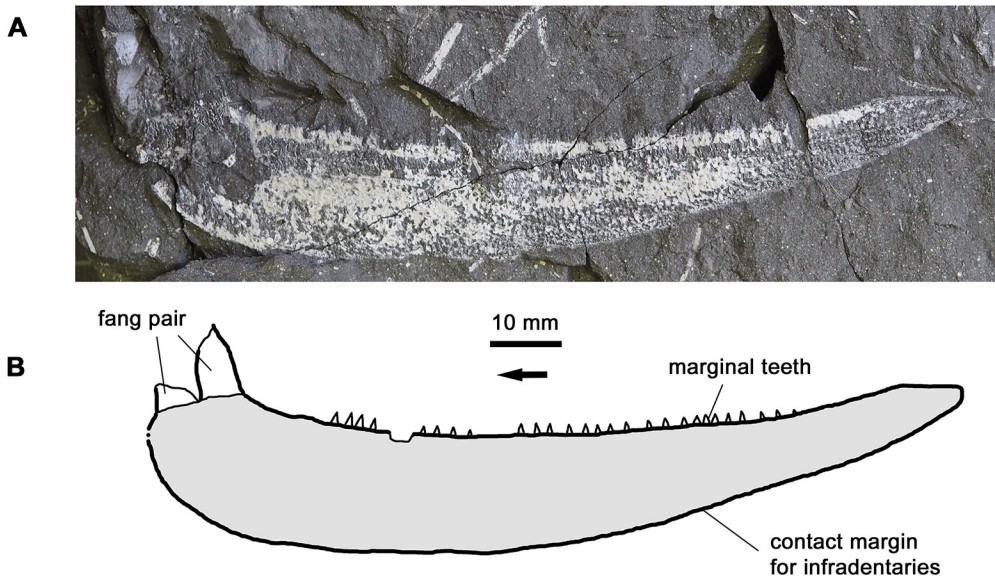

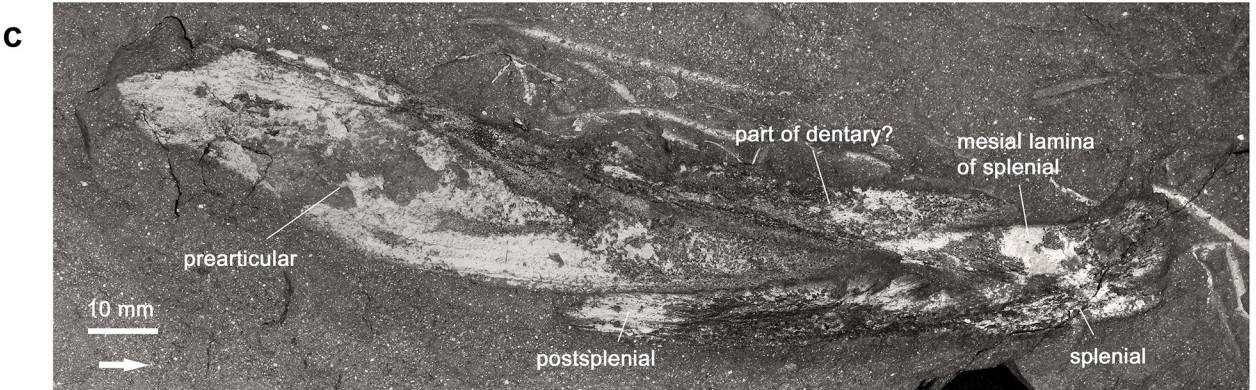

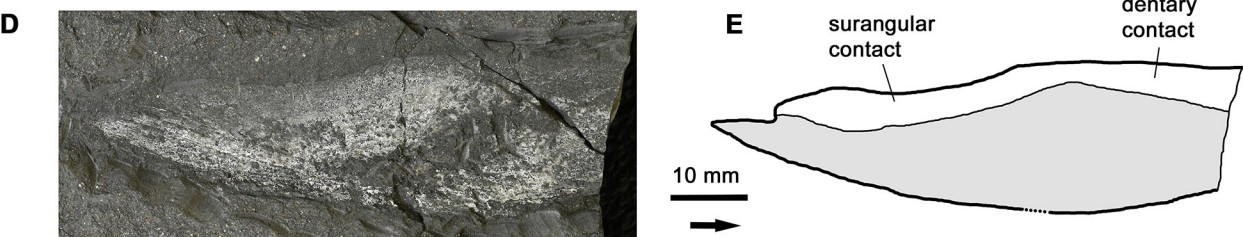

**Fig 6. The lower jaw.** A-B, photo and interpretative drawing of dentary AM6502a. C, labelled photo of lower jaw ramus AM18000a. D-E, A-B, photo and interpretative drawing of angular AM5668. Graphic conventions as in Fig 4.

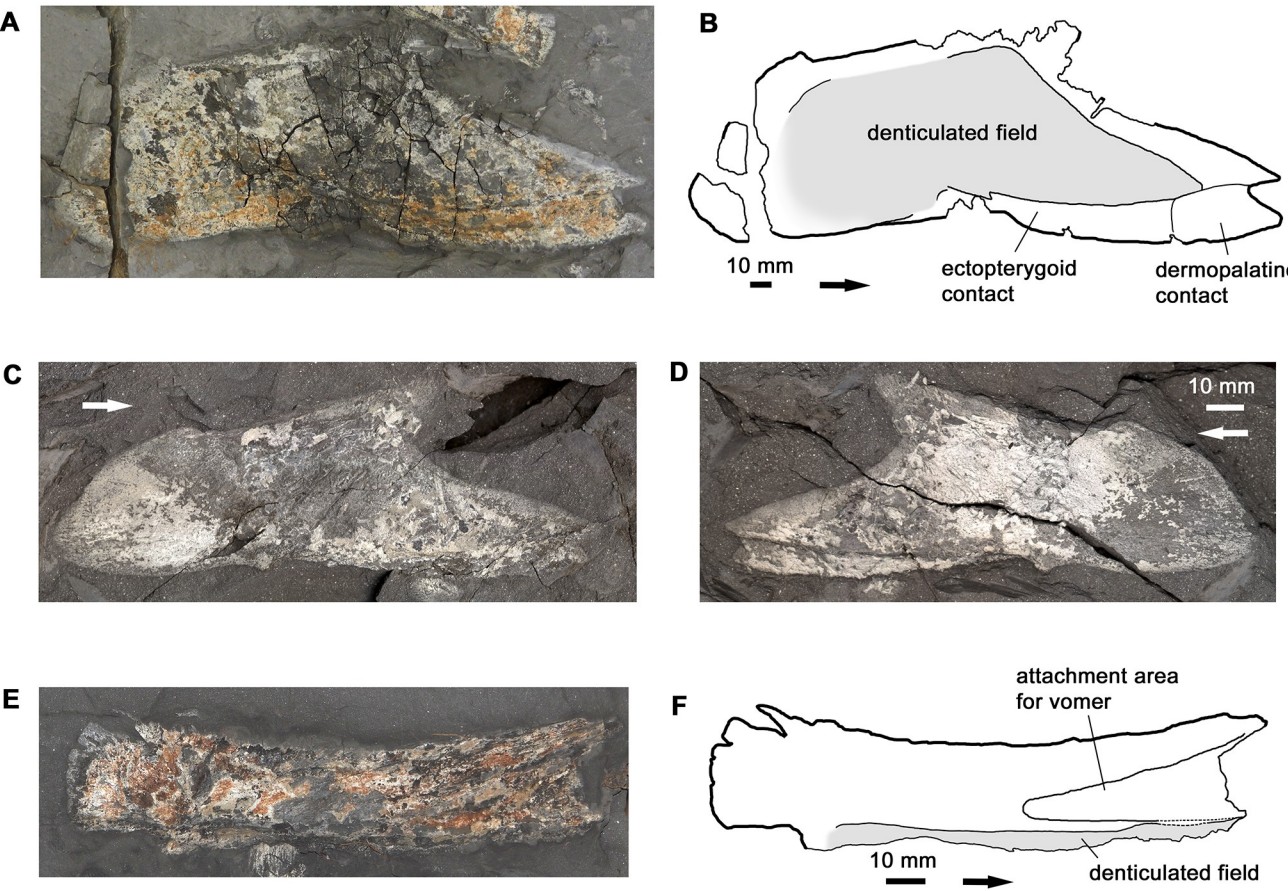

**Fig 7. Palatal bones.** A-B, photo and interpretative drawing of entopterygoid AM6528a. B also incorporates information from AM6528b. C-D, photos of entopterygoid AM6508b+a, part and counterpart. E-F, photo and interpretative drawing of parasphenoid AM6528a. Graphic conventions as in Fig 4.

**Operculogular series.** The bony gill cover is represented by specimens of the opercular, subopercular and principal gular (Fig 8). No definite submandibulars have been recovered, although a rectangular bone with a large overlap area at one end (tentatively interpreted as a supracleithrum) may in fact come from this series. The most noteworthy aspect of the operculogular series is the great depth of the opercular and the correspondingly shallow subopercular. The opercular of *H. udlezinye* is nearly three times as deep as the subopercular, whereas in *Eusthenopteron* it is less than twice as deep [22]. In *Hyneria lindae* the opercular is incompletely known, but the subopercular has a very different shape to that of *H. udlezinye* with a much deeper posteroventral region [17: fig 8C,D]. As the subopercular of *H. udlezinye* is represented by several specimens of different sizes, all with the same distinctive shallow shape, we are confident in the taxonomic significance of this difference. The principal gular is of generalized tristichopterid type.

**Pectoral girdle.** Only the dermal elements of the pectoral girdle are preserved (Fig 9). These include definite examples of the lateral extrascapular, anocleithrum, cleithrum and clavicle. The lateral extrascapular has a smoothly rounded posterior margin. If its size relative to the postparietal shield has been inferred correctly, the median extrascapular was rather wide. The anocleithrum is very similar to that of *Hyneria lindae* [17]. Its ornamented area is triangular as in *H. lindae*, *Edenopteron* and *Mandageria* [4, 6, 17], contrasting with the four-sided ornamented area of *Eusthenopteron* [22: fig 126]. The cleithrum shows a degree of individual

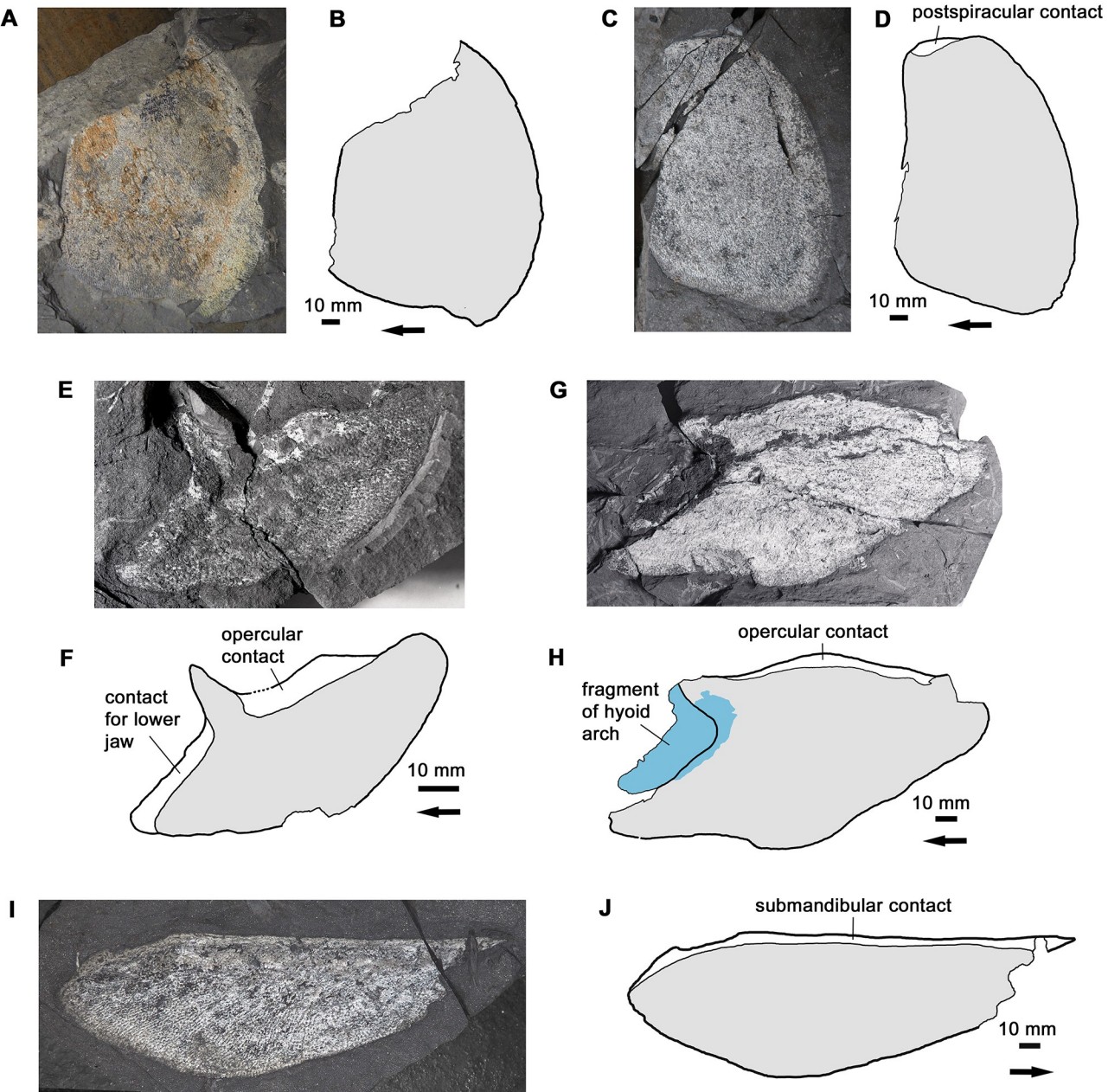

**Fig 8. The gill cover.** A-B, photo and interpretative drawing of opercular AM5532. C-D, photo and interpretative drawing of opercular AM6521a. E-F, photo and interpretative drawing of subopercular AM5249. G-H, photo and interpretative drawing of subopercular AM5391bi. I-J, photo and interpretative drawing of gular AM5888. Graphic conventions as in Fig 4. In H, blue indicates a partially preserved hyoid arch element adhering to the subopercular.

variation in shape (Fig 10). As is typical for tristichopterids, the contact margin for the clavicle is concave. A distinctive feature shared with *H. lindae* is the strongly developed and finger-like anteroventral extension.

**Ulnare.** The only bone to be preserved from the appendicular skeleton is an ulnare (Fig 11A and 11B). It carries a large postaxial flange and is very similar to the corresponding element in *Eusthenopteron* [22: fig. 102].

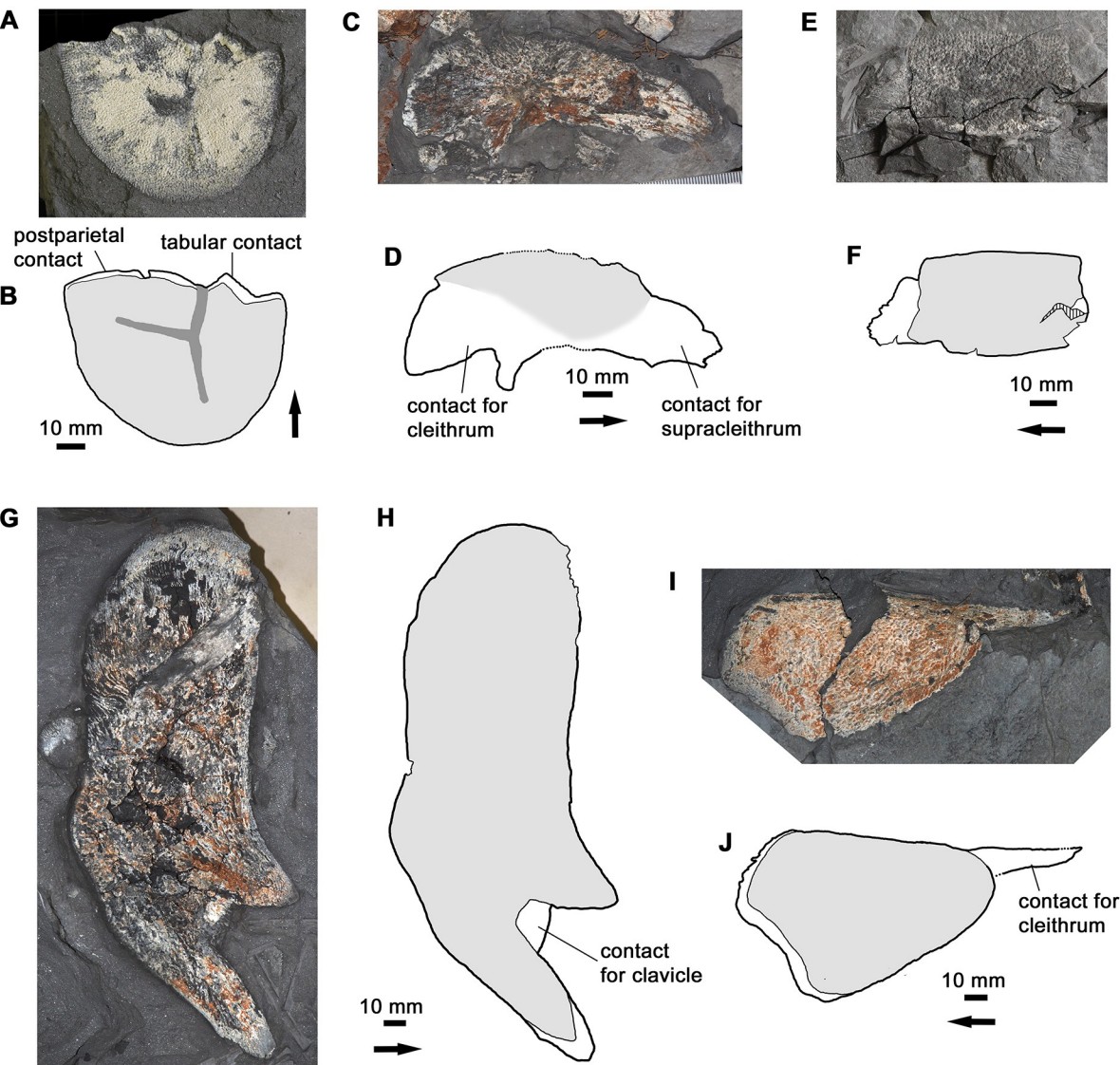

**Fig 9. The pectoral girdle.** A-B, photo and interpretative drawing of lateral extrascapular, AM6542b. B also includes information from counterpart. C-D, photo and interpretative drawing of anocleithrum, AM6528a. E-F, photo and interpretative drawing of possible supracleithrum (or submandibular), AM18003. G-H, photo and interpretative drawing of cleithrum, AM6540b. I-J, photo and interpretative drawing of clavicle, AM6528b. J also includes information from counterpart. Graphic conventions as in Fig 4.

**Axial skeleton.** The only preserved part of the axial skeleton, apart from isolated bones, is a partly articulated series of arches with fin radials from the posterior part of the vertebral column, associated with a triangular basal plate from either the anal or the posterior dorsal fin (Fig 11C). Interestingly, the shape of the basal plate resembles that of non-tristichopterids such as the porolepiform *Glyptolepis* [24, 25] rather than *Eusthenopteron* [22]. The arches comprise one evidently articulated series and a scatter of disrupted elements; we surmise that the articulated elements are most probably neural arches, held in place by the body musculature after death, whereas the scattered elements are haemal arches that have been displaced by the rupture of the body cavity. If this is correct, it implies that the neural arches of the tail region articulated distally with radials of the epichordal lobe of the caudal fin, as in *Cabonnichthys* [3],

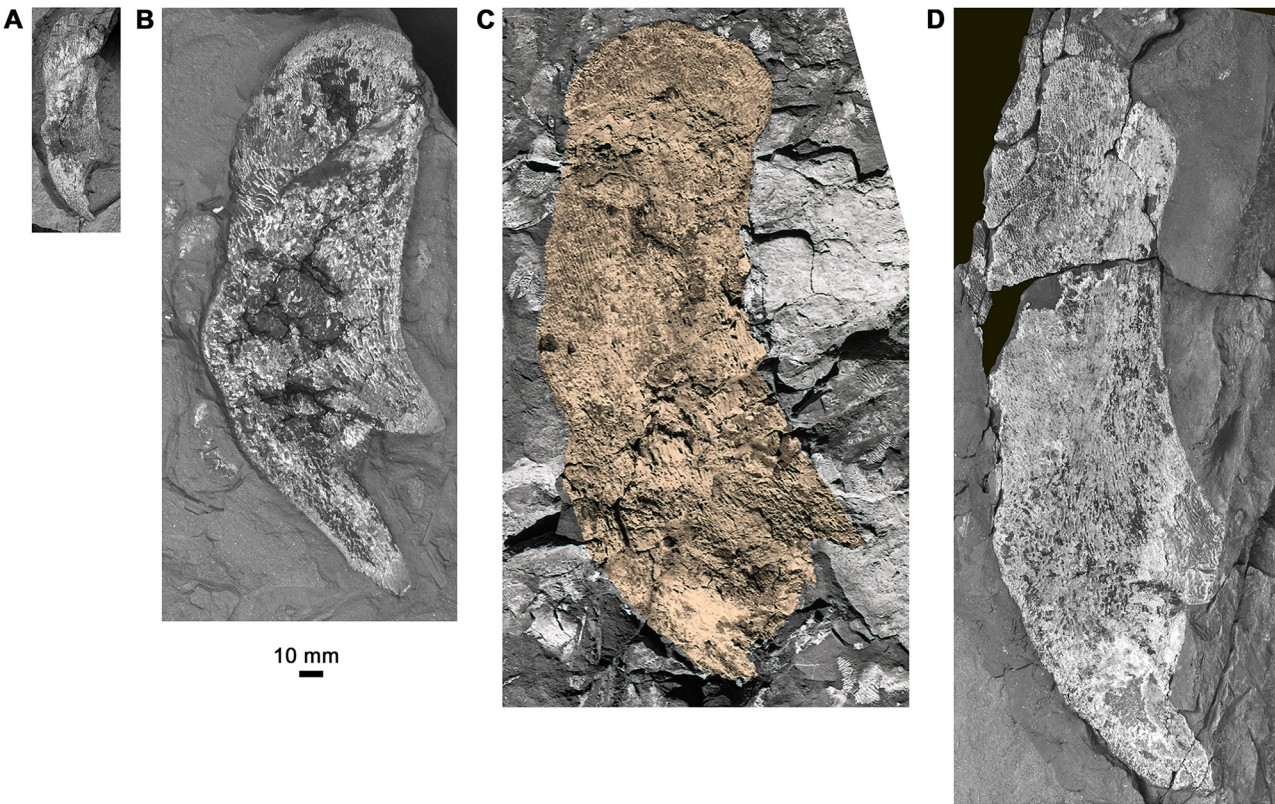

**Fig 10. Variability of the cleithrum.** A, cleithrum 6516a,b (composite image of part and counterpart). B, cleithrum AM6540b. C, cleithrum AM5389a, highlighted in gold to facilitate visualisation. D, cleithrum AM6545. All to same scale.

contrasting with the condition in *Eusthenopteron* where such radials are absent [22]. Vertebral centra are not preserved and were presumably cartilaginous.

**Squamation and lepidotrichia.**   Numerous isolated scales and some areas of lepidotrichia are preserved in association with the dermal bones of *Hyneria udlezinye* (Fig 12). The scales are characterised by an extensive, deeply folded margin of the trailing edge, a character uniquely shared with *Hyneria lindae* (Fig 12B). Isolated, elongated scales with an extremely extended trailing edge (Fig 12C–12E), up to 75 mm in length, are interpreted as probable basal scutes from the fins [22].

## Discussion

It is a straightforward matter to identify the largest sarcopterygian from Waterloo Farm as a giant tristichopterid. All the characteristics of the preserved bones fall squarely within the general gestalt and range of variation typical of these fishes. Assuming body proportions similar to those of *Mandageria*, the only large tristichopterid for which it has been possible to assemble a whole-body reconstruction [4], the total body length of the holotype individual of *Hyneria udlezinye* was in the range of 180–190 cm; some isolated bones, however, come from larger individuals. An isolated cleithrum (AM 6545), for example, is 50 percent larger than that of the type specimen (36 cm from dorsal to ventral extremity as opposed to 24 cm), suggesting a possible body length of at least 270 cm for this individual (Fig 10D). This size range is similar to other giant tristichopterids.

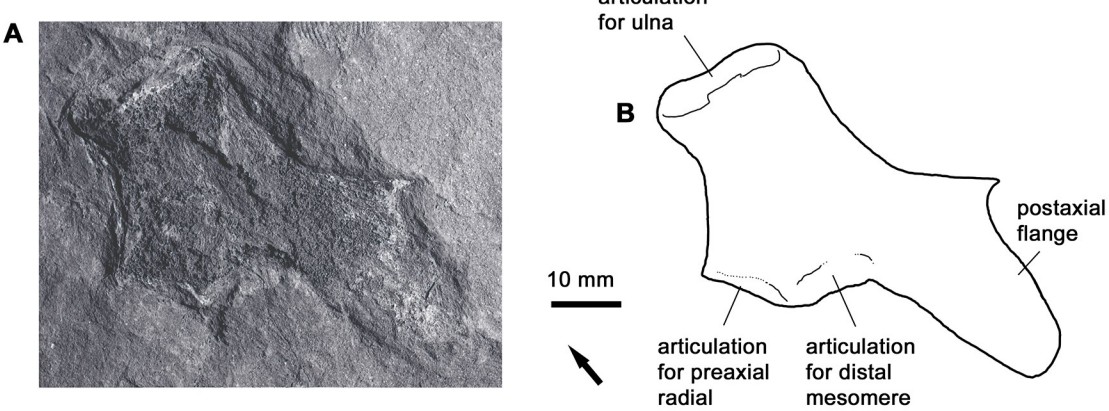

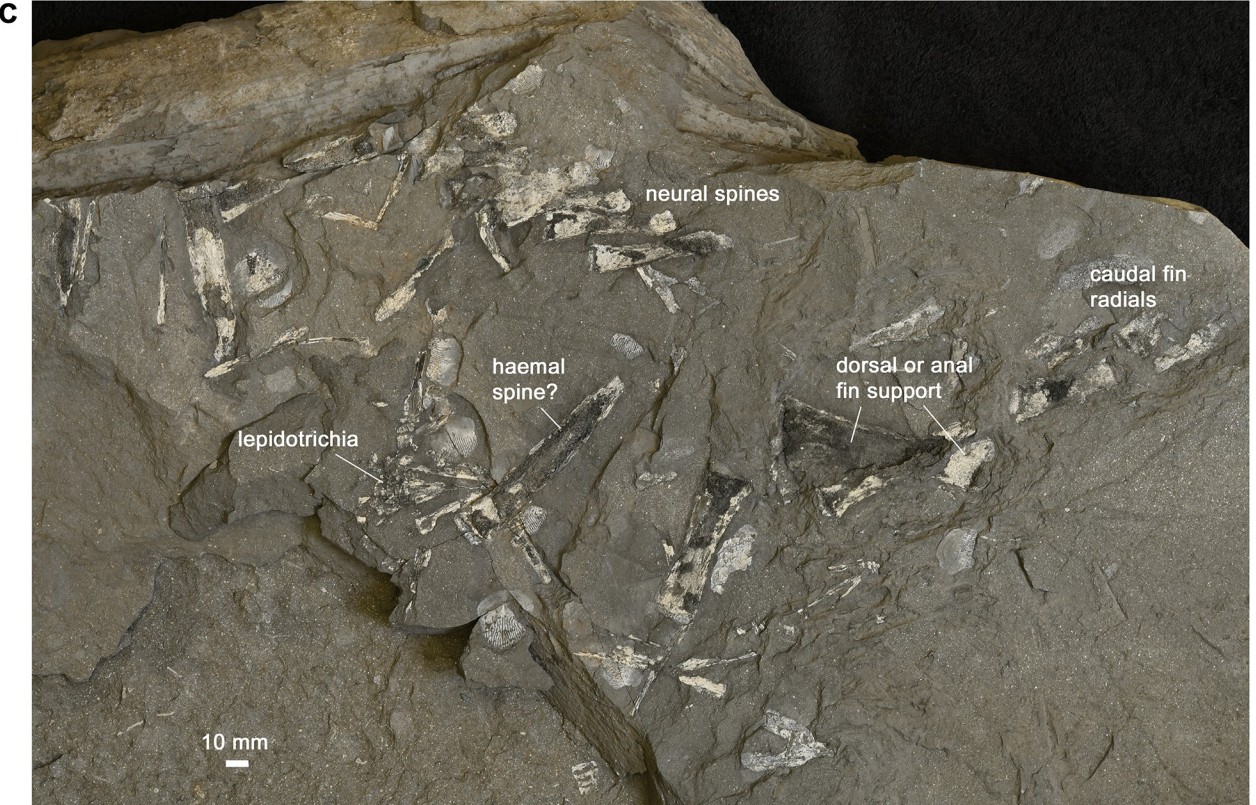

**Fig 11. Postcranial endoskeleton.** A-B, photo and interpretative drawing of ulnare AM4868. C, photo of scattered axial elements from the posterior part of the body, AM18002. Graphic conventions as in Fig 4.

A precise identification is more challenging, because the diversity of these top predators is surprisingly high: even if we restrict the analysis to known Famennian taxa we have to consider the genera *Langlieria*, *Eusthenodon*, *Edenopteron* and *Hyneria* [2]. The features that assign the Waterloo Farm material to *Hyneria* are the broad parietal shield with its distinctively short postpineal region, the ornament (which is finer than in *Eusthenodon* and *Edenopteron*) and the morphology of the scales, which are uniquely characterized by an extensive, deeply folded

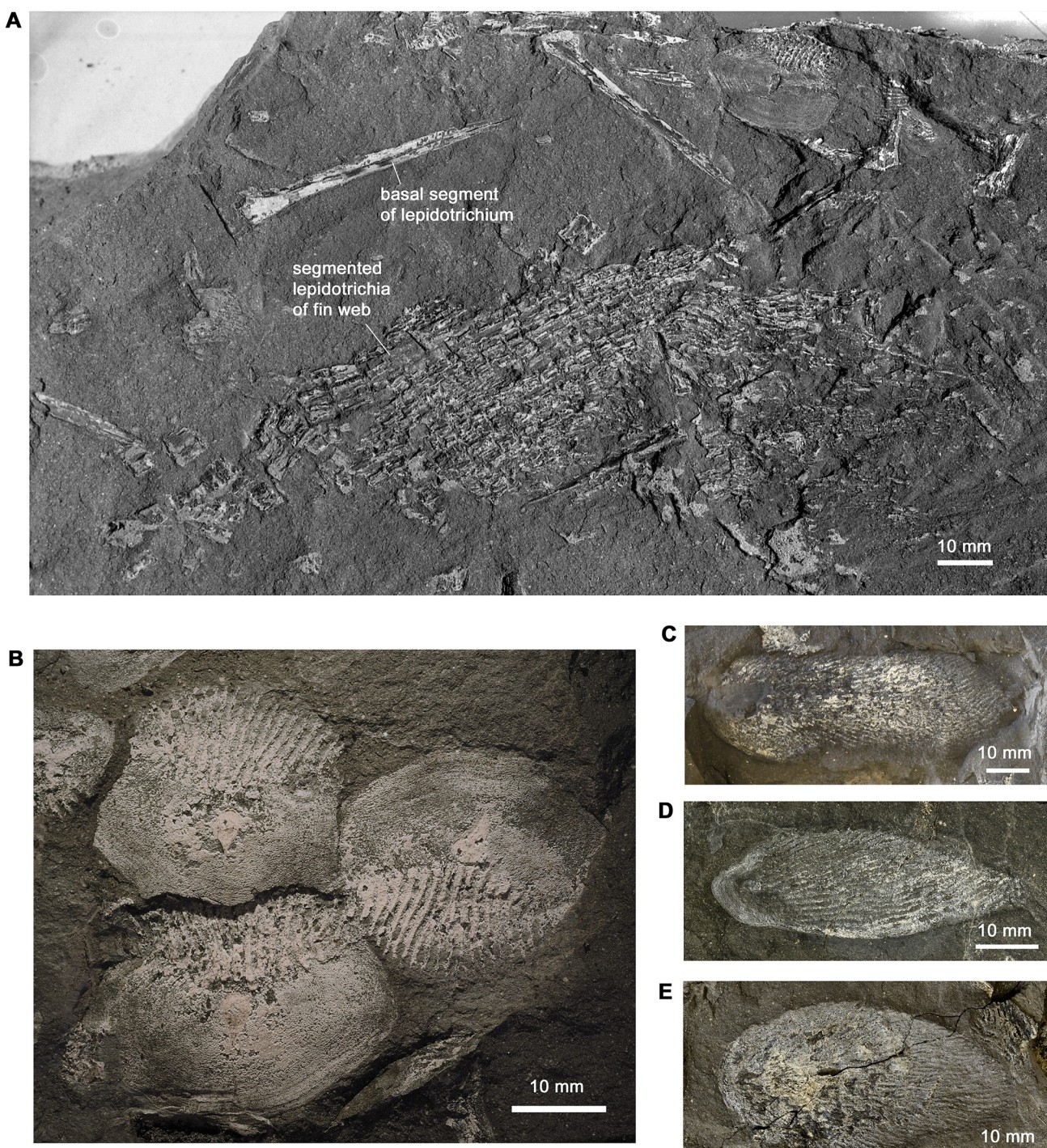

**Fig 12. Lepidotrichia and scales.** A, slab containing fin web fragment with articulated lepidotrichia, together with displaced basal segments of lepidotrichia and scales, AM5389ei. B, slab showing three typical body scales, AM18004. C-E, probable basal scutes of fins, C, AM6540. D, AM5389. E, AM6533b.

margin to the trailing edge. *Hyneria udlezinye* differs from *H. lindae*, the type species, in a number of respects. The proportions of the prearticular and lacrimal, and the morphology of the maxilla, appear less derived and closer to general conditions for the Tristichopteridae. Conversely, the shallow subopercular and exceptionally large (though incompletely preserved) dentary fangs of *H. udlezinye* appear to be autapomorphic.

Comparison with the published reconstructions of other giant tristichopterids suggests that *Hyneria udlezinye* has an unusually large branchial chamber. A measure of caution is needed, however, because the branchial region is difficult to reconstruct accurately as the bones frequently become displaced or pushed over each other. The gill cover of *Eusthenodon*, reconstructed by Jarvik [5], certainly seems implausibly small. On the other hand, *Mandageria* and *Cabonnichthys* have been reconstructed with gill covers approximately similar in size to each other and *Eusthenopteron*, all of them slightly smaller than *H. udlezinye* [3, 4, 22]. *Hyneria lindae* appears to have gill cover bones of similar size to *H. udlezinye*, although the subopercular is proportionately deeper, but the skull has never been reconstructed [17]. We conclude from this that *Hyneria* probably has a larger branchial chamber than other known tristichopterids, although the exact magnitude of the difference is uncertain. This may reflect life in oxygen-poor waters, or perhaps a very static lifestyle where the respiratory current was not significantly supplemented by forward movement through the water.

The discovery of *Hyneria udlezinye* increases the known diversity of large, late tristichopterids, and presents us with the top non-chondrichthyan predator of the Waterloo Farm ecosystem (Fig 13). However, arguably its main importance lies in the light that it casts on tristichopterid (and more generally, vertebrate) biogeography of the Late Devonian. Giant tristichopterids are globally distributed in the Famennian and appear to be almost ubiquitous members of non-marine to brackish ecosystems. They occur from the tropics of Euramerica (for example in East Greenland and at Andreyevka-2 in central Russia), through the southern mid-latitudes (the Catskill Formation of Pennsylvania and the Worange Point Formation of New South Wales), down to the Antarctic Circle (Waterloo Farm). Some genera were widespread. *Eusthenodon* is recorded from Greenland, Belgium, central Russia and Australia [2, 8, 9]; *Langlieria* from Belgium and Pennsylvania [26]; and now *Hyneria* from Pennsylvania and South Africa. In general terms this speaks to the interconnectedness of the Famennian world, but on a more specific level it would be interesting to know where this radiation of giants originated.

The recent study by Olive *et al.* [2] addressed this question by applying Bayesian Binary MCMC analysis to the geographic distribution of known tristichopterids, mapped onto a parsimony tree. The analysis suggested that the *Langlieria* lineage may have originated in western Europe, whereas the top-end clade containing *Cabonnichthys*, *Mandageria*, *Eusthenodon*, *Edenopteron* and *Hyneria* is interpreted as most probably an Australian radiation. The result undoubtedly reflects the fact that *Cabonnichthys* and *Mandageria* are known only from the Frasnian locality of Canowindra, and *Edenopteron* only from the Famennian Worange Point Formation, both in New South Wales. Much of Gondwana is poorly sampled for Late Devonian vertebrates, potentially skewing the results; it should be noted in this context that the Givetian to Frasnian Aztec Siltstone of Antarctica contains not only the rather primitive giant tristichopterid *Notorhizodon*, but also fragmentary remains of possibly more derived *Eusthenodon*-like forms [29]. It may thus be that the apparent Australian biogeographical signature detected by Olive *et al.* [2] is actually a broader Gondwanan signature partly obscured by patchy sampling. The discovery of *Hyneria udlezinye* fits very nicely into this context by showing that *Hyneria* also has a Gondwanan distribution. Indeed, the great distance (both in kilometers and palaeolatitude) of Waterloo Farm from the Catskill Formation suggests that *Hyneria* may have been widely distributed in Gondwana.

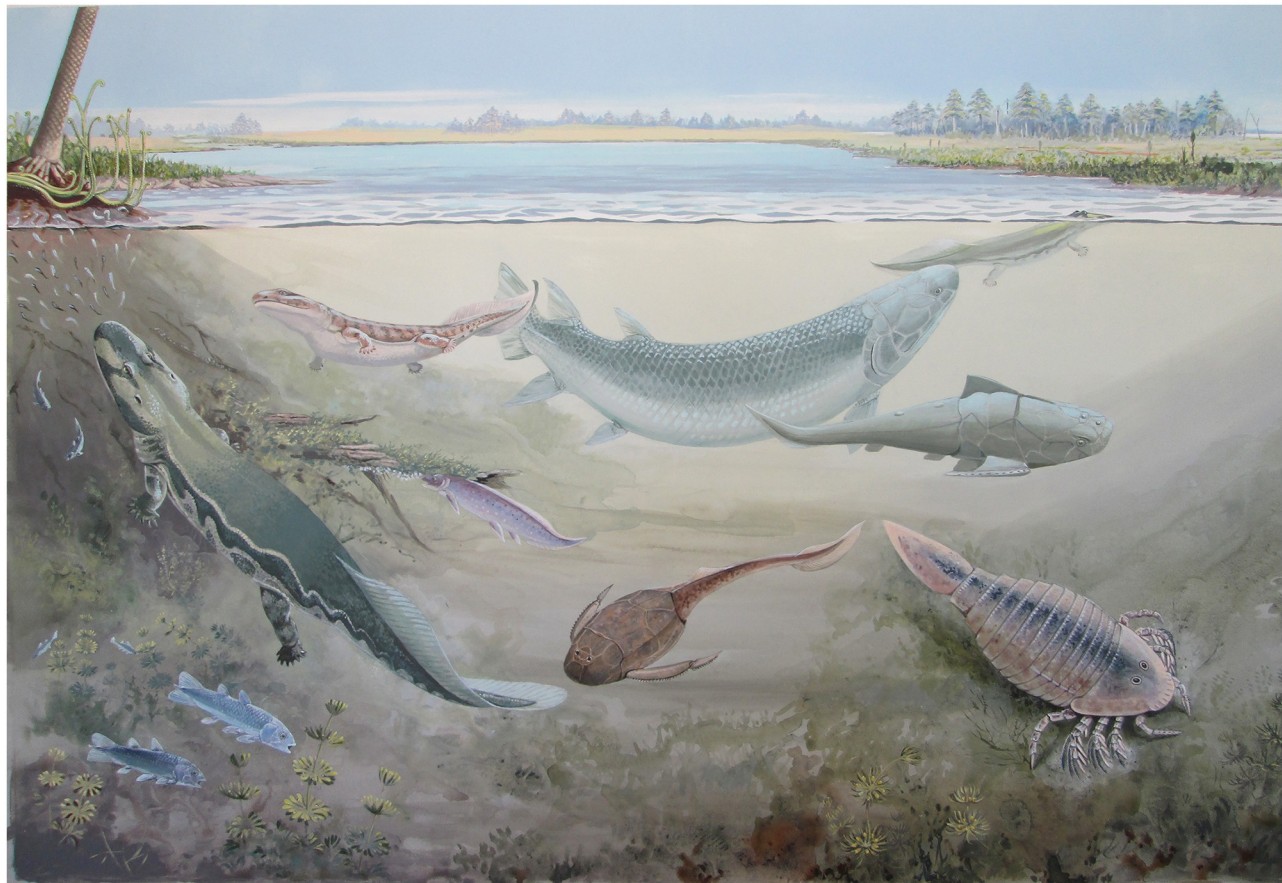

**Fig 13. Life reconstruction of the non-marine component of the Waterloo Farm biota.** *Hyneria udlezinye* is shown together with the tetrapods *Umzantsia amazana* and *Tutusius umlambo* [11], the placoderms *Groenlandaspis riniensis* and *Bothriolepis africana* [19], the coelacanth *Serenichthys kowiensis* [27], the lungfish *Isityumzi mlomomde* [28], and a cyrtoctenid eurypterid. Painting by Maggie Newman, copyright R. W. Gess.

## Conclusion

The largest osteichthyan member of the Waterloo Farm vertebrate assemblage, a predatory sarcopterygian with a probable maximum length of nearly three metres, proves to be a new species of the genus *Hyneria*. This genus is otherwise only recorded from the late Famennian Catskill Formation of Pennsylvania. The new species, *Hyneria udlezinye*, differs from the type species *Hyneria lindae* in a number of minor but securely attested proportional characters relating to the skull roof, cheek, lower jaw and operculum. *Hyneria* now joins *Eusthenodon* and *Langlieria* as one of the derived, late, giant tristichopterids known from both Euramerica and Gondwana. The other confirmed members of this clade (*Mandageria*, *Cabonnichthys* and *Edenopteron*) are exclusively known from Gondwana. This strongly supports the contention that this clade represents a Gondwanan radiation [2].

*Hyneria udlezinye* is the first tristichopterid to be recorded from a high palaeolatitude, all other members of the group coming from palaeoequatorial to mid-palaeolatitude localities. All previously recorded Gondwanan members of the derived tristichopterid clade come from Australia, leading Olive *et al*. to argue for an Australian origin for this clade [2]. The new evidence from Waterloo Farm, however, suggests that a more general Gondwanan origin for this clade is highly likely. This once again demonstrates how inferences about biogeographical patterns have historically been skewed by a paucity of data from high-palaeolatitude localities. Such

data can only come from Gondwana, as no continents extended into northern high latitudes during the Devonian. The Waterloo Farm lagerstätte provides a unique window into an almost unknown part of the Late Devonian world.

## Supporting information

**S1 File.**
(PDF)

**S2 File.**
(PDF)

**S1 Questionnaire.**
(DOCX)

## Acknowledgments

RG acknowledges valuable discussions with M. I. Coates (University of Chicago) regarding much of the material described herein. The South African National Road Agency Limited (SANRAL) is thanked for their assistance in 1999 and 2005 with the rescue of palaeontological material from Waterloo Farm during roadworks. Ryan Nel is thanked for lab assistance and assistance with photography. Shawn Johnstone is thanked for assistance with mounting of the type specimen.

## Author Contributions

**Conceptualization:** Robert W. Gess, Per E. Ahlberg.

**Data curation:** Robert W. Gess.

**Funding acquisition:** Robert W. Gess, Per E. Ahlberg.

**Investigation:** Robert W. Gess, Per E. Ahlberg.

**Methodology:** Robert W. Gess, Per E. Ahlberg.

**Project administration:** Robert W. Gess.

**Validation:** Robert W. Gess, Per E. Ahlberg.

**Visualization:** Robert W. Gess, Per E. Ahlberg.

**Writing – original draft:** Robert W. Gess, Per E. Ahlberg.

**Writing – review & editing:** Robert W. Gess, Per E. Ahlberg.

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
