## [Decision Letter · Decision Letter 0]

8 Dec 2022

PONE-D-22-29333A high latitude Gondwanan species of the Late Devonian tristichopterid Hyneria (Osteichthyes: Sarcopterygii)PLOS ONE

Dear Dr. Ahlberg,

Thank you for submitting your manuscript to PLOS ONE. After careful consideration, we feel that it has merit but does not fully meet PLOS ONE’s publication criteria as it currently stands. Therefore, we invite you to submit a revised version of the manuscript that addresses the points raised during the review process.

Dear Dr. Ahlberg,

your manuscript has been evaluated by two referees. While one suggested acceptance as the manuscript is, the other had two minor comments. I would like to ask you to integrate these comments.

Besides, I would like you to consider the Recommendation 11A of the International Code of Zoological Nomenclature which states that "An unmodified vernacular word should not be used as a scientific name. Appropriate latinization is the preferred means of formation of names from vernacular words." therefore the specific part of the new name should be latinized. Otherwise, in the explanation ot the name etymology please state that this is an apposition.

Yours sincerely,

Mikołaj Zapalski

We look forward to receiving your revised manuscript.

Kind regards,

Mikołaj K. Zapalski, Ph. D., D. Sc.

Academic Editor

PLOS ONE

Journal Requirements:

"RG acknowledges valuable discussions with M. I. Coates (University of Chicago) regarding much of the material described herein. The South African National Road Agency Limited (SANRAL) is thanked for their assistance in 1999 and 2005 with the rescue of palaeontological material from Waterloo Farm during roadworks. Ryan Nel is thanked for lab assistance and assistance with photography. Shawn Johnstone is thanked for assistance with mounting of the type specimen. RG is funded by the Millennium Trust, GENUS (DSI-NRF Centre of Excellence in Palaeosciences) and the NRF of South Africa. PEA acknowledges the support of a Wallenberg Scholarship from the Knut & Alice Wallenberg Foundation and an Advanced Grant (ERC-2020-ADG 10101963 "Tetrapod Origin") from the European Research Council."

 "PEA: Wallenberg Scholarship (not numbered), from the Knut & Alice Wallenberg Foundation. https://kaw.wallenberg.org

PEA: ERC Advanced Grant ERC-2020-ADG 10101963 "Tetrapod Origin", from the European Research Council. https://erc.europa.eu/homepage

RWG: Millennium Trust, South Africa (no number). https://www.mtrust.co.za

RWG: GENUS (DSI-NRF Centre of Excellence in Palaeosciences), South Africa (no number). https://www.genus.africa

RWG: NRF of South Africa (no number). https://www.nrf.ac.za

Please include your amended statements within your cover letter; we will change the online submission form on your behalf.54. Please include your full ethics statement in the ‘Methods’ section of your manuscript file. In your statement, please include the full name of the IRB or ethics committee who approved or waived your study, as well as whether or not you obtained informed written or verbal consent. If consent was waived for your study, please include this information in your statement as well. 

a) You may seek permission from the original copyright holder of Figure 1 to publish the content specifically under the CC BY 4.0 license.  

6. We note that Figure 13 in your submission contain copyrighted images. All PLOS content is published under the Creative Commons Attribution License (CC BY 4.0), which means that the manuscript, images, and Supporting Information files will be freely available online, and any third party is permitted to access, download, copy, distribute, and use these materials in any way, even commercially, with proper attribution. For more information, see our copyright guidelines: http://journals.plos.org/plosone/s/licenses-and-copyright.

a) You may seek permission from the original copyright holder of Figure 13 to publish the content specifically under the CC BY 4.0 license. 

8.  Please take this opportunity to be sure you have met all of our guidelines for new species. For proper registration of a new zoological taxon, we require two specific statements to be included in your manuscript.

 1.) In the Results section, the globally unique identifier (GUID), currently in the form of a Life Science Identifier (LSID), should be listed under the new species name, for example: Anochetus boltoni Fisher sp. nov. urn:lsid:zoobank.org:act:B6C072CF-1CA6-40C7-8396-534E91EF7FBBAnother LSID for the manuscript itself should also appear within the Nomenclature statement. You will need to contact Zoobank (zoobank.org/About) to obtain a GUID (LSID). You should receive one LSID for your manuscript and a separate, unique LSID for the new species.  2)
Please also insert the following text into the Methods section, in a sub-section to be called "Nomenclatural Acts": The electronic edition of this article conforms to the requirements of the amended International Code of Zoological Nomenclature, and hence the new names contained herein are available under that Code from the electronic edition of this article. This published work and the nomenclatural acts it contains have been registered in ZooBank, the online registration system for the ICZN. The ZooBank LSIDs (Life Science Identifiers) can be resolved and the associated information viewed through any standard web browser by appending the LSID to the prefix "" ext-link-type="uri" xlink:type="simple">http://zoobank.org/". The LSID for this publication is: urn:lsid:zoobank.org:pub: XXXXXXX. The electronic edition of this work was published in a journal with an ISSN, and has been archived and is available from the following digital repositories: PubMed Central, LOCKSS [author to insert any additional repositories]. All PLOS ONE articles are deposited in PubMed Central and LOCKSS. If your institute, or those of your co-authors, has its own repository, we recommend that you also deposit the published online article there and include the name in your article.Following a recent ruling by the International Commission on Zoological Nomenclature, electronic journals are now a valid format for publication of new zoological taxa. In order to ensure the valid publication of your new species, please be sure to include the updated version of Nomenclatural Acts (above). A complete explanation of our guidelines for publishing new species can be found on our website: http://www.plosone.org/static/guidelines#zoological.

Additional Editor Comments:

Dear Dr. Ahlberg,

your manuscript has been evaluated by two referees. While one suggested acceptance as the manuscript is, the other had two minor comments. I would like to ask you to integrate these comments.

Besides, I would like you to consider the Recommendation 11A of the International Code of Zoological Nomenclature which states that "An unmodified vernacular word should not be used as a scientific name. Appropriate latinization is the preferred means of formation of names from vernacular words." therefore the specific part of the new name should be latinized. Otherwise, in the explanation ot the name etymology please state that this is an apposition.

Yours sincerely,

Mikołaj Zapalski

Reviewers' comments:

Reviewer's Responses to Questions

**Comments to the Author**

1. Is the manuscript technically sound, and do the data support the conclusions?

Reviewer #1: Yes

Reviewer #2: Yes

2. Has the statistical analysis been performed appropriately and rigorously? 

Reviewer #1: N/A

Reviewer #2: N/A

3. Have the authors made all data underlying the findings in their manuscript fully available?

Reviewer #1: Yes

Reviewer #2: Yes

4. Is the manuscript presented in an intelligible fashion and written in standard English?

Reviewer #1: Yes

Reviewer #2: Yes

5. Review Comments to the Author

Reviewer #1: I have no major issues with this manuscript, just two clarification questions about the criteria used to establish the fossil examined as a new species.

1. On lines 167-168, the authors state that the specimen looks very similar to an existing type species, with only "proportional differences in some bones" differentiating them. Would it be possible to illustrate that these proportional differences are NOT simply a difference in size? In other words, show that the proportional differences are not always in the same direction between the species.

2. The authors also note that the teeth in this new fossil are "much less well preserved than the bone" and that even the largest "do not show complete natural outlines". Given this, I question using the size of the dentary fangs as a diagnostic character for the species. Could the authors provide more support for this particular diagnostic character?

Reviewer #2: This is a well written paper based on a reasonably good material of a giant tristichopterid Hyneria (Sarcopterygii) from the late Famennian of Waterloo Farm in South Africa. The major strength of this work is the detailed description of a new specimen which is similar to the type species, H. lindae, but with more completely preserved anterior part. Particularly, the presence of skull bones adds much new information to our previous knowledge on this genus. The material, isolated bones, are well illustrated and interpreted. As I am not an expert on sarcopterygians I cannot judge if the proposition of giving a new species name (H. udlezinye) to the specimen under description, despite its general similarity to H. lindae, is justified. However, as the new specimen comes from Gondwana, and the older material from Laurussia (Pennsylvania), such a solution seems acceptable. H. udlezinye is not the first giant tristichopterid found from Gondwana, but the first one which comes from high palaeolatitudes.

Generally, although I do not think this paper brings ground-breaking new information, I consider it useful and important, and I am sure it deserves publication.

6. PLOS authors have the option to publish the peer review history of their article (what does this mean?). If published, this will include your full peer review and any attached files.

Reviewer #1: No

Reviewer #2: **Yes: **Michał Ginter

---

## [Author Response · Author response to Decision Letter 0]

21 Dec 2022

We thank the reviewers and academic editor for their constructive comments. Here are our responses to both:

Specific name: We note the ICZN recommendation but do not entirely agree with it. Linnaeus himself named the silver bream (a cyprinid fish) Abramis bjoerkna, using the Swedish vernacular name "björkna" as the specific name. Furthermore, there is a strong current trend for using unmodified vernacular names: Tiktaalik (Inuktituk) and Guiyu (Mandarin Chinese) are familiar recent examples. We will stick with udlezinye but have noted in the text that it is an apposition.

Style requirements: We have applied the style templates in the links provided by you.

Inclusivity questionnaire: Completed and uploaded.

Funding information / Financial Disclosure: We have removed the funding-related text from the Acknowledgements. The online Financial Disclosure statement is correct. In the Current Funding Sources List, the GENUS award to Rob Gess is missing because this funding initiative (which falls under the National Research Foundation) cannot be found in your data base. 

Ethics statement: We have included an ethics statement in the Methods section.

Figure 1 (maps): Figure 1A has previously been published in PLoS ONE (Gess Trinajstic 2017, doi: 10.1371/journal.pone.0173169) and should thus not present a problem. Figure 1B is reproduced from Gess Ahlberg 2018 (Science 360, doi: 10.1126/science.aaq1645). Science's copyright policy (uploaded) states specifically that the author retains copyright and is free to re-use the image without restrictions as long as it is attributed. The figure legend has been modified to cite the sources of the images.

Figure 13: The copyright of this image (a painting by Maggie Newman) resides with the first author, who commissioned it from her. A letter explaining its status and granting you permission to reproduce the image has been uploaded. The figure legend has been modified to state the copyright.

Registration in ZooBank: This has been carried out and the new LSIDs have been added to the manuscript in accordance with your instructions.

Comments of Reviewer #1:

1) It is evident for two reasons that the shape differences we observe between Hyneria lindae and H. udlezinye are taxonomically significant rather than just allometric size effects. Firstly, large individuals of the two species are of approximately the same size. Secondly, in the case of the subopercular (arguably the most morphologically distinctive bone of H. udlezinye), small and large examples show the same dorsoventrally shallow shape, as demonstrated in our figure 8E-F and G-H (compare the scale bars!). We have amended the text to emphasise that these suboperculars are similar in shape even though they differ in size.

2) The preservation of the dentary fangs (and other large teeth) in the H. udlezinye material is really interesting: they are typically black, whereas the bone is silver, and their outlines are often 'wavy' as well as incomplete. It looks like they were completely demineralised with only the organic matrix of the dentine being preserved. However, we are confident that the dentary fangs are larger than in H. lindae, for the simple reason that the preserved parts are of noticeably larger dimensions. The only uncertainty concerns exactly how tall they were, but there is no positive reason to believe that their proportions were different from those of H. lindae or many other Devonian-Carboniferous sarcopterygians. We have not amended the text.

Comments of Reviewer #2:

We appreciate this positive assessment of the value of our paper.

---

## [Editor Report · Decision Letter 1]

23 Jan 2023

A high latitude Gondwanan species of the Late Devonian tristichopterid Hyneria (Osteichthyes: Sarcopterygii)

PONE-D-22-29333R1

Dear Dr. Ahlberg,

We’re pleased to inform you that your manuscript has been judged scientifically suitable for publication and will be formally accepted for publication once it meets all outstanding technical requirements.

Kind regards,

Mikołaj K. Zapalski, Ph. D., D. Sc.

Academic Editor

PLOS ONE

Additional Editor Comments (optional):

Dear Prof. Ahlberg,

Thank you for introducing changes in the text, and, where relevant, explaining convincingly why the changes are not desired. The manuscript can be accepted in its present form.

My best wishes in 2023!

Mikołaj Zapalski
---

## [Editor Report · Acceptance letter]

26 Jan 2023

PONE-D-22-29333R1 

A high latitude Gondwanan species of the Late Devonian tristichopterid *Hyneria* (Osteichthyes: Sarcopterygii) 

Dear Dr. Ahlberg:

I'm pleased to inform you that your manuscript has been deemed suitable for publication in PLOS ONE. Congratulations! Your manuscript is now with our production department. 

Kind regards, 

on behalf of

Dr. Mikołaj K. Zapalski 

Academic Editor

PLOS ONE